



# Geo-climatic hazards in the eastern subtropical Andes: Distribution, Climate Drivers and Trends

Iván Vergara[1], Stella M. Moreiras[2, 3], Diego Araneo[2, 3] and René Garreaud[4, 5]

[1] CONICET-IPATEC, Bariloche, 8400, Argentina
[2] CONICET-IANIGLA, Mendoza, 5500, Argentina
[3] National University of Cuyo, Mendoza, 5502, Argentina
[4] University of Chile, Santiago, 8330015, Chile
[5] Center for Climate and Resilience Research, Santiago, 8320198, Chile

*Correspondence to*: Iván Vergara (ivergara@comahue-conicet.gob.ar)

**Abstract.** Detection and understanding of historical changes in the frequency of geo-climatic hazards (G-CHs) is crucial for the quantification of current hazard and their future projection. Here we focus in the eastern subtropical Andes (32-33° S), using meteorological data and a century-long inventory on 553 G-CHs triggered by rainfall or snowfall. First we analysed their spatio-temporal distributions and the role of climate variability on the year-to-year changes in the number of days with
G-CHs. Precipitation is positively correlated with the number of G-CHs across the region and year-round; mean temperature is negatively correlated with snowfall-driven hazards in the western (higher) half of the study region during winter, and with rainfall-driven hazards in the eastern zone during summer. The trends of the G-CHs frequency since the mid-20th century were calculated taking cautions for their non-systematic monitoring. The G-CHs series for the different triggers, zones and seasons were generally stationary. Nonetheless, there is a small positive trend in rainfall-driven G-CHs in the eastern zone
during summer congruent with a rainfall increase there. We also found a decrease in snowfall-driven G-CHs in the western zone since the late 1990's onwards, most likely due to a reduction in winter precipitation rather than an increase in temperature.

## 1 Introduction

Geo-climatic hazards are natural phenomena that occur by a combination of atmospheric (e.g., precipitation, temperature,
wind) and terrain factors (geotechnical and morphometric properties). This definition includes landslides, snow avalanches and phenomena of glacial (surges, GLOFs, IDLOFs) and fluvial origin (floods, lateral erosions, avulsions). All of them constitute an important risk along the Argentinean-Chilean Andes that runs for nearly 4000 km along the west part of South America, especially in its central portion (32-33° S) that has the highest erosion rates (Carretier et al., 2013), reaches more 5 km in elevation and where geo-climatic hazards have caused considerable human and economic losses mostly to the
international traffic crossing this sector (e.g., Sepúlveda and Moreiras, 2013; Moreiras et al., 2018).



There is some work about historical and projected trends in the frequency of regional geo-climatic hazards (G-CHs; e.g., Moreiras, 2006; Moreiras and Vergara, 2017) but a quantitative analysis has not yet been carried out in order to attribute their origin. A first issue to elucidate is the origin of a seemingly increase in the number of G-CHs triggered by rainfall since the mid-20th century (Moreiras and Vergara, 2017), which could be due to an actual change in precipitation or to an increase

in monitoring efforts during the last decades. A second question is on the origin of a decrease in the frequency of G-CHs triggered by snowfall since the end of the 1990s (Moreiras and Vergara, 2017). This could be due to a decrease in winter precipitation, an increase in the freezing level during these hazards due to the ongoing regional warming or both (Masiokas et al., 2006; Vuille et al., 2015).

The purpose of this research is to assess the significance of the G-CHs trends and investigate their causes. To attribute these

trends, we began by improving our general understanding of regional G-CHs analysing their distributions in time and space, and establishing their relationship with different climate elements. This research took advantage of a long, updated record of G-CHs in the eastern side of the subtropical Andes, with daily resolution and precise spatial location along a portion of an international, highly transited highway connecting Chile and Argentina. On the other hand, meteorological information is rather poor in this area, with few surface stations and absence of other measurement systems (radar, local radiosondes, etc.).

The paper is organized as follows. We begin by describing the physical characteristics of the area, the assembly of the G-CHs record and its usage. In the results section we present the spatio-temporal characterization of the G-CHs, followed by their association with the local climate and the temporary changes of the climatic variables and the G-CHs. In the discussion section, the results obtained are analysed, a future research direction is proposed, and a conceptual evaluation of the future geo-climatic hazard of the region is carried out. Finally, in the conclusions section, the most important results are described.

## 1.1 Study area and geographical setting

At subtropical latitudes (32-33° S) the Andes cordillera separates central Chile (to the west) and western central Argentina (to the east) with the border approximately following the highest peaks of the range, that reaches over 5 km ASL in this sector (Fig. 1). Here, an international road and the ex-Trasandino railway, links the cities of Los Andes (Chile) and Mendoza (Argentina). This is a major commercial and touristic route in use since the 17th century connecting the east and west side of

the continent. Our work relies on the historical G-CHs record taken in the Argentinean side of the road and railway, extending eastward from the border down to the plains near the city of Mendoza. This sector coincides with the middle and upper Mendoza river basin (Fig. 1a) and hosts more than 10 thousand inhabitants. Anthropic changes in the landscape are negligible due to the absence of significant cultivated areas and the very limited infrastructure. The National Route 7 has currently an average traffic of about 3000 vehicles per day (ONDaT, 2018), maintained throughout the year due to domestic

travels and the daily opening of the international pass, except when large snowfalls or major G-CHs occur.

The study area has elevations from 1250 to 5970 m ASL (buffer of 15 km with respect to the Mendoza River; Fig. 1b). The lower limit of discontinuous permafrost is at 3700 m ASL approximately (Trombotto et al., 1997). The area covered by perennial snow and glaciers (including inactive rock glaciers) is 103 km$^2$ (2.7 % of the total area; IANIGLA, 2018). The


study area encompasses the morphotectonic units Cordillera Principal, Cordillera Frontal and Precordillera (Ramos 1996), so

the geotechnical, geological and morphometric characteristics vary widely. The first morphotectonic units are strongly altered by Quaternary glacial activity, with U-shaped valleys that have highly steep walls.

In the ravine headwaters there is a large amount of unconsolidated material generated by the seismic activity, the latest glaciations and the current cryoclasticism. Considering this factor and the dry climate of the region (described below), it is likely that the frequency and magnitude of debris flows are limited by the availability of water and not by debris supply. The

vegetation types are Andean steppe and Monte scrubland (Paruelo et al., 2001) and their low density favours the triggering of landslides and snow avalanches. Debris flows are triggered by shallow failure planes and/or in-channel entrainment (Mergili et al., 2012) with little or no influence of the edaphic humidity generated by previous rainfall (Vergara et al., 2018). Through photointerpretation and field surveys, Moreiras (2009) mapped a total of 869 landslides in the middle and upper Mendoza river basin. Based on that work, the abundance of the different types of landslides was established: debris flows 79

%, falls 9 %, rotational and translational slides 7 %, and complex landslides 5 %. More recently, Moreiras et al., (2012) used historical sources to describe 72 landslides and snow avalanches in the upper Mendoza river basin in the period 1822-2010, allowing an identification of their triggering causal factors (henceforth triggers or drivers): total precipitation 77 %, snowmelt 14 % and seismicity 9 %. The most important conditioning factors of landslides in the middle Mendoza river basin are lithology and slope (Moreiras 2005a).

**1.2 Climate**

The subtropical Andes, with its impressive altitude (>5000 m ASL), continuity and nearly north-south orientation, acts as a barrier separating two distinct climate regimes (Garreaud, 2009; Viale et al., 2019). The western side of the Andes receives most of the precipitation during austral winter (May-September) due to the arrival of cold fronts and other disturbances moving from the Pacific ocean (Falvey and Garreaud, 2007; Viale and Nuñez, 2011). In this side, frontal precipitation tends

to be homogeneous (encompassing hundreds to thousands of kilometres) but tends to increase with altitude causing a vertical gradient in accumulation from about 300 mm yr$^{-1}$ in the Chilean lowlands to about 1000 mm yr$^{-1}$ atop of the Andes (e.g., Viale and Garreaud, 2015). The freezing level during winter storms is around 2500 m ASL (Garreaud, 2013) so that a significant portion of the precipitation over the western Andes is in the form of snow. Although the strong westerly flow atop of the Andes can spill over some snow towards its eastern side, the Argentinean sector receive most of the precipitation

during austral summer (October to March; Viale and Garreaud, 2014) in connection with the southern edge of the South American Monsoon (e.g., Vera et al, 2006). In sharp contrast with the west side, precipitation to the east has a convective nature and its water source is ultimately the Atlantic Ocean. These precipitation systems usually have an extension of 10 km$^2$ and intensities that can double those of the frontal systems (raw data of Vergara et al., 2018). The convection presents the maximum peak of probability during evening and early night, and minimum in the morning (Saluzzi, 1983).

In our study area (Fig. 1a), the western (wintertime frontal) and eastern (summertime convective) regimes have varying degrees of influences in the zonal direction (recall that the Andes axis is north-south oriented so moving in longitude the





terrain altitude changes rapidly). Of course, there is a continuous change in the nature and seasonality of precipitation across the region (e.g., Fig. 2a) but a division into a western (W) an eastern (E) zones is useful for subsequent analyses. The mean elevation of W and E zones are 3740 and 2700 m ASL, respectively (Fig. 1b), and both have a similar zonal lengths (between 43.6 and 46.1 km). The mean climate features were obtained by averaging the stations data available in each zone for the period 1993-2017 (Table 1). Because of the marked vertical gradients and the different features of valley sections where the stations are located, the monthly averages at each station were transformed into standardized anomalies before being averaged. The eastern (E), lower zone has a semi-arid climate with summer dominated, convective precipitation (Araneo et al., 2011; Fig. 2c). The climate of the western (W), higher zone is of the Tundra type (Sarricolea et al., 2016) with frontal precipitation concentrated during austral winter (Fig. 2b-c). Note that both sectors share the same annual cycle of temperature, with warm summers and cold winters (Fig. 2d).

The regional precipitation has a high inter-annual standard deviation, about 35-40 % with respect to the annual average (Garreaud et al., 2009). El Niño Southern Oscillation (ENSO) is the major driver of these changes in the western sector / western side of the Andes, with a clear tendency for above (below) normal winter precipitation during El Niño (La Niña) years (see review in Garreaud et al., 2009). Indeed, in the study area there is a positive correlation between the ENSO and the number of landslides and snow avalanches (Moreiras, 2005b, Moreiras et al., 2012). The Pacific Decadal Oscillation (PDO) and the Antarctic Oscillation (AAO) also modulate Andean precipitation. The PDO increase precipitation during its positive phases, while AAO does so during its negative phase (e.g., Masiokas et al., 2006). These oscillations have a greater impact on the wintertime precipitation. El Niño (La Niña) years also tend to produce above (below) normal precipitation in the eastern half of our study region, although the ENSO impact on the monsoonal regime at subtropical latitudes is rather weak (e.g., Montecinos et al., 2000) and other large-scale modes also contribute to interannual variability in summer (e.g., Scian et al., 2005).

## 2 Data and methods

Our starting point for the present analysis is the historical record of landslides in the middle Mendoza river basin for the period 1790-2003 (Moreiras, 2006) as well as the landslides and snow avalanches inventory in the upper Mendoza river basin for the period 1822-2010 (Moreiras et al., 2012). The recorded G-CHs are concentrated in ravines, talus cones and rock walls adjacent to the valleys talwegs of the Mendoza and Uspallata rivers, where the routes and the railway are located, and includes several types of G-CHs: debris flow, mud flow, hyper-concentrated flow, debris avalanche, rock fall, debris fall and snow avalanche (Fig. 3). For the 18th and 19th centuries, these records are mainly composed by travellers' notes where the location and date of the hazards were usually indicated. For the 20th century, these records are mainly composed by articles from regional newspapers and written communications from railway and road companies. These sources indicate the date and position where the routes or the railway were blocked, and sometimes also inform the trigger, time and description of the main deposit. For the 21st century, field surveys of scientists from the Argentine Institute of Nivology and Glaciology were





added, so the record became more detailed and the number of landslides and snow avalanches increased considerably. In

addition to this, we continue using regional newspapers (El Andino, Los Andes Online, MDZ, Los Andes, Diario Uno, El

Sol) as well as printed communications from Argentinean and Chilean customs and communications from road institutions

(National Highway Management and Provincial Highway Management).

Snowfall and rainfall are the only triggers that we study here. Historically, G-CHs triggered by snowfall events have

produced much more fatalities than those triggered by rainfall (7 vs. 50, Fig. 4c). The G-CHs triggered by glacial and fluvial

phenomena or wind and rapid melting of snow and/or ice within the active layer were excluded. For each landslide and snow

avalanche in the historical record, the most probable trigger was inferred from the original sources (the reports indicate if

there were rain, snow, earthquakes, etc.) or were established using in-situ or satellite data. To decide on the occurrence of

daily precipitation, we used 13 stations in the study area (Table 1) complemented by satellite estimates from the CMORPH

daily product (Joyce et al., 2004; since 12/2002). The freezing level in the days with G-CHs was estimated using daily mean

temperature data at 8 surface stations (Table 1) from where the Zero Isotherm Altitude (ZIA) was calculated employing a

wet adiabatic gradient of 6 °C km$^{-1}$. The freezing level was obtained subtracting 125 m from the ZIA to account the average

distance that take for snow crystals to become liquid (Garreaud, 1992, White et al., 2010). Finally, to discard (or confirm)

seismic-driven hazards we employed seismicity reports (USGS, 2018).

In total, 553 G-CHs were collected for the period 1882-2017 (Fig. 4). There were 55 additional landslides which were

discarded because the trigger could not be established, and another 35 G-CHs were discarded because it was not possible to

recover the precise geographic location. Since in many G-CHs the magnitude was not available we have refrained from

using this variable.

Rainfall exclusively triggers landslides while snowfall can trigger snow avalanches and landslides. Indeed, about 35% of the

snowfall-driven hazards in the W zone (high elevation sector) and the 83 % in the E zone (lower elevation sector) were rock

or debris falls. During the snowfall these landslides can be triggered by short intervals of rain or by the melting of snow

when it is deposited on the rock (and the consequent interstitial pressure in the fractures). The snow avalanches were

considered to be triggered by snowfall, although they may be occur some weeks afterwards, due to rains or an increase in

temperature and radiation. This simplification was done because the anomalous snowfall is the main atmospheric cause of its

occurrence.

The greatest deficiency of the record is its generation from a non-systematic monitoring, depending on whether there was an

observer that recorded a G-CH. To avoid to the maximum this error, most of the analyses were carried out for the period

1961-2017 (unless otherwise indicated). In 1961, the last section of international Route 7 was paved leading to a stabilization

of traffic and G-CHs reports.

Using the G-CHs record, two time series were made. The first is the sum of all G-CHs during a given period (season or year)

referred to as the *number of G-CHs*. The second is the sum, over a given period, of the number of days in which there was at

least on G-CH, referred to as the *number of days*. The *number of G-CHs* is more representative of the spatial extent and

intensity of the meteorological event but has a larger bias by non-systematic monitoring. We therefore use the *number of*



*days* (with at least one G-CH) per season as our primary variable for analysis, segregating by trigger (snowfall, rainfall) and zone (W, E).

## 3 Results

### 3.1 Spatio-temporal distributions of G-CHs

Figure 5b show the probability of G-CHs annual occurrence for each ravine, talus cone and rock wall within our study region. As expected, snowfall-induced hazards are concentrated in the upper, western zone, although a few ones occur in the eastern, lower sector. Likewise, rainfall-induced landslides concentrate in the E zone but also extends into the W zone. The rainfall trigger has the 4 highest values (maximum probability 67 %) and the snowfall trigger the 5th and 8th highest values (maximum probability 40 %; Fig. 5b). To obtain a more complete view of the G-CHs spatio-temporal distribution we calculated the annual probability density of these hazards across the full study area (following the longitudinal axis) using the non-parametric kernel density estimator for directional-linear data (García-Portugués et al., 2017). The bandwidth was selected through the maximum likelihood cross-validation (Hall et al., 1987). The snowfall-driven hazards presented probability densities greater than the rainfall-driven landslides with a well-defined peak in W zone during May-June (early winter) but extending to the full winter semester (Fig. 5a). The probability densities of the rainfall-driven hazards are more diffuse within the E zone and maximize at the height of the austral summer (Jan-Feb). Overall, the probability of G-CHs closely follows the annual cycle of the precipitation (Fig. 2a-c).

The snowfall-driven hazards are concentrated in the W because here the winter precipitation is greater and in solid state due to elevation. On the other hand, rainfall triggered landslides is greater in the E zone due to the lower elevation and the higher occurrence of intense, convective rainfall over this area during summer. While the snowfall trigger can only occur in the wintertime, due to the need for a low ZIA, the rainfall trigger can occur throughout the year, due to the fact that a sector of the E zone is below the mean freezing level of winter precipitation (Fig. 5b). The small peaks of activity of the E zone at the west boundary in winter and at the east boundary end in winter-spring coincide with the precipitation of the two sub-zones of the Atlantic domain (Fig. 2a-c).

### 3.2 Association with climate drivers

We now establish the association between G-CHs with climate drivers (precipitation and temperature) at interannual time scales, considering the different combinations of seasons (winter, summer) and zones (W, E). Due to the non-normality in the precipitation and G-CHs series, the Spearman correlation coefficient was used. The significances of the correlations were evaluated with the method proposed by Zar (1972) considering a 95 % level of confidence (same percentage used in the following cases). Precipitation exhibits a positive correlation in all combinations, so that above (below) average precipitation tend to increase (decrease) G-CHs in both zones throughout the year (Fig. 6a). Winter precipitation is significantly correlated with snowfall-induced hazards across the whole study region, while summer precipitation is significantly correlated with





rainfall-induced landslides in the East zone only (recall that summer rainfall in the West zone is low). The larger correlation
values during winter in the W zone may results from the spatially coherent pattern of frontal precipitation, in contrast with
the isolated, convective nature of summer precipitation. Another reason may be that the G-CHs of this spatio-temporal
combination are generally snow avalanches which increase their occurrence probability with precipitation accumulation. On
the contrary the falls and flows, most common in summer and E zone, depend on sub-daily precipitation intensity, data that
is not available in this region.

Correlation values with average temperature are lower than those with precipitation and differ in sign among zones and
seasons (Fig. 6b). The temperature is negatively and significant correlated with snowfall-driven hazards in the West zone
during winter, and with rainfall-driven hazards in the East zone during summer. The weaker G-CHs–Temperature
correlations may reflect the lack of a direct relationship between these variables, but rather an indirect association mediated
by precipitation.

**3.3 Contemporaneous changes**

The linear trend over the period 1961 to 2017 for the precipitation and the G-CHs is calculated for each zone and season,
using the non-parametric Mann-Kendall statistic (Mann, 1945, Kendall, 1975). The linear slope was estimated with the non-
parametric statistic developed by Sen (1968). These methods are considered more adequate than least squares approach
given the non-normality of the series. In the higher W zone, precipitation exhibit non-significant trends that differs between
seasons: negative in winter (the wet season) and positive in summer (Table 2 and Fig. 7a,e). The winter drying trend is in
line with the negative rainfall tendency observed along the lowlands of central Chile (Boisier et al., 2016; 2018) that has
been accentuated in the last decade in connection with the so-called central Chile mega drought (Garreaud et al., 2017;
Rivera et al., 2017). Instead, in the E zone, precipitation has increased year round but only significantly during winter (Table
2 and Fig. 7b,f). These results coincide with the significant increase of precipitation in the central-western argentine plain
that occurred mainly between December and May (SADSN, 2018; Labraga, 2010; Vera and Díaz, 2015). The trends in the E
and W zones remain similar when considering the full period 1957-2017 (Table 2).

Given their episodic nature, finding trends in the G-CHs frequency is more difficult. Nonetheless, rainfall-driven G-CHs
show a significant increase in their number during summer over the E zone that is congruent with the weak precipitation
increase in this sector and season (Fig. 7 b,d and Table 2).

The snowfall-driven winter hazards exhibit a decrease in the W zone since the end of the 1990s [-0.7 d with G-CHs decade$^{-1}$
for the 1997-2017 period (p<0.05); Fig. 7g]. This decrease may be connected with the precipitation decline or with an
elevation of the freezing level due to the ongoing regional warming (Masiokas et al., 2006; Vuille et al., 2015). To assess the
role of the temperature, we examine its evolution in the May-September season during the 1974-2017 period. To this end, we
used surface temperature data from two nearby stations in the Chilean side at a high elevation (Lagunitas and Embalse el
Yeso; Table 1). For dry days we calculated the ZIA using a dry adiabatic gradient of 6.5 °C km$^{-1}$ and for days with
precipitation we calculated the freezing level (see Sect. 2).



The linear trends were calculated for the resulting ZIA series, through the least squares method. The ZIA for dry days has a significant trend of +4.31 m yr$^{-1}$ (Fig. 8) in agreement with previous studies (Carrasco et al., 2008) and the overall warming over the subtropical Andes (Falvey and Garreaud, 2009; Vuille et al., 2015; Vergara et al., 2019). On the other hand, the

freezing level for days with precipitation had a very weak, non-significant trend of +0.42 m yr$^{-1}$ (Fig. 8), similar to the trend obtained by Carrascto et al., (2005) in the annual ZIA for days with precipitation and notably lower than the trend during dry days. This suggest that the warming over the last decades has -for now- limited impact in the decrease in snowfall-driven G-CHs.

As an independent method, elevations of the main deposits of the snowfall-driven G-CHs for the periods 1882-1953 (39

cases) and 1993-2015 (52 cases) were compared (Fig. 9). The significance of difference in mean elevations between both periods was determined through the non-parametric Bootstrap method (999 simulations; Zieffler et al., 2011). The mean elevations of main deposits of the snowfall-driven G-CHs of both periods did not have a significant difference. These results lend support to the notion that the decrease in snowfall-driven winter hazards is largely caused by the decline in Pacific-sourced rainfall rather than an elevation in the ZIA due to regional warming.

**4 Discussion**

The seasonal mean temperature exhibits a weak and non-systematic correlation with the *number of days* with G-CHs, reflecting the lack of a direct physical link between these variables. By contrast precipitation accumulated during winter is positively associated with the *number of days* with snowfall-driven hazards in both zones, while summer rainfall is correlated with the *number of days* with rainfall-driven landslides in the East zone. Although some of these correlations are

statistically significant, their values are not high enough to fit predictive models at inter-annual (Pavlova et al., 2014) or meteorological event scale (e.g., Staley et al., 2017). An improved, denser meteorological network across this complex terrain (including radar monitoring of summer convection) may result in a better depiction of the climate control of the local geo-climatic hazard, suitable for a more quantitative diagnosis, forecast and projections of their future frequency.

Our trend analysis indicates that winter precipitation has been decreasing in the W zone. Although barely significant, this

decrease is associated with a more robust decline in frontal, Pacific-sourced precipitation in central Chile (Boisier et al., 2016) and the occurrence of a decade-long drought in that region (Garreaud et al., 2017). On the other hand, we found an overall increase in precipitation in the E zone. The trend is more significant in the winter semester due to the enhanced moisture transport from the Atlantic to the east of the Andes (SADSN, 2018; Barros et al., 2014; Vera and Díaz, 2015). We also studied the winter temperature given their control on altitude separating rainfall and snowfall (in close correspondence

with ZIA). In agreement with other evidence of regional warming, we found a clear increase in ZIA during dry days, which are the majority in this region. In the subset of days with precipitation, however, the ZIA exhibits an insignificant increase, suggesting that the thermal structure of winter, Pacific sourced storms hasn't changed enough yet to cause an impact on the number of snowfall-driven G-CHs over the subtropical Andes.





Considering the interannual correlation between G-CHs and climate elements, as well as the tendencies of the later, we
advanced in the attribution of the trends in the *number of days* with snow avalanches and landslides. Finding trends in the G-
CHs frequency is complex given their highly variable nature and the non-systematic monitoring, resulting mostly in non-
significant values. Nonetheless, two series of G-CHs show significant trends. The first is the increasing *number of days* with
rainfall-driven hazards during summer over the E zone, in line with the weak precipitation increase in this sector and season.
Secondly, we confirm a decrease in the *number of days* with snowfall-driven hazards during winter in the W zone since the
end of the 1990s. We further show that such decrease is consistent with the decline in winter precipitation over central Chile
and the adjacent Andes, with little or no effect of the regional warming in the last decades since the freezing level during
precipitating days exhibit an insignificant change.

When corrections for non-systematic monitoring are not applied (variable used, type and start year of trends) much larger
tendencies are obtained. For example, when the trend of rainfall trigger, E zone and summer season G-CHs is calculated
using *number of G-CHs* variable, the least squares method and the 1950-2017 period, a tendency of 1.5 G-CHs decade$^{-1}$ is
obtained, instead of 0.2 d with G-CHs decade$^{-1}$ (consider that on average in a day with G-CHs there are 2.2 hazards; Figure
4b). This indicates that the previously found increase in the record of rainfall-driven G-CHs since the mid-20th century
(Moreiras and Vergara, 2017) was mainly due to increases in monitoring.

A weakness of this study is that it does not account for the magnitude of landslides and snow avalanches. A future research
direction, that may include this intensity metric, could be to stablish a relationship between the G-CHs and fluvial solid
discharge, the latter showing to be a regionally good proxy of rainfalls (Garreaud and Viale, 2014). Positive results could add
a magnitude proxy to the existing series and increase their reliability.

In closing this section, we speculate on the future prospect of geo-climatic activity over our study area. Model-based climate
projections (e.g., Junquas et al., 2012; Vera and Díaz, 2015; Bozkurt et al., 2018) consistently reveal (a) a marked warming
over the subtropical Andes, (b) a decline of Pacific-sourced, winter precipitation, and (c) an increase in Atlantic-sourced,
monsoonal precipitation during the warm season. The last two projections, seemingly acting in the present, will result in the
maintenance of the observed trends during the next decades: an increase in rain-driven hazards in the E zone and a decrease
in snowfall-driven hazards in the W zone, eventually amplified by the projected warming.

## 5 Conclusions

In this work we have analysed a long record of G-CHs (including geo-location and possible trigger) in a sector extending
from the subtropical Andean crest toward the lowlands of western Argentina (city of Mendoza) that follows a 90 km long
transect of a highly used international highway. We focused our work on landslides and snow avalanches that can be
triggered by rainfall and snowfall. The purpose of the research was to calculate the trends of the G-CHs and to know the
causes of the changes found in the G-CHs frequency. The precautions taken before calculating trends were to separate
spatially and temporally climate regimes and avoid as much as possible the effects of non-systematic monitoring. The G-



CHs series for the different triggers, zones and seasons were generally stationary, however, two series had tendencies. The rainfall-driven G-CHs during summer over the E zone had an increase since the mid-20th century, which was related to the weak rainfall increase in that zone and season. The calculated increase is much lower than that which would have been obtained without corrections for non-systematic monitoring. On the other hand, snowfall-driven G-CHs in the W zone had a

decrease since the late 1990's onwards. In this case, non-systematic monitoring cannot explain the trend since it only increased over time. This change was attributed to a decrease in winter precipitation, since the snowline increase occurred mainly in the days without precipitation.

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

**Author contribution:** Contributed to conception and design: IV, SM, DA, RG. Contributed to acquisition of data: IV, SM. Contributed to analysis and interpretation of data: IV, SM, DA, RG. Drafted the article: IV, RG.

**Acknowledgments:** We are grateful to Dr. Eduardo García-Portgués for the advice in the kernel density estimation for directional-linear data.

**Competing interests:** The authors declare that they have no conflict of interest.





**Table 1:** Stations used and their main characteristics. The registration period does not consider interruptions. The stations without asterisks were used for triggers determination and statistical analysis, and those with one (two) asterisks only for triggers determination (statistical analysis).

| Station | Institution | Elevation (m ASL) | Coordinates (° S and W) | Record (yr) Precipitation | Temperature |
|---|---|---|---|---|---|
| West zone | | | | | |
| Riecillos | DGA | 1290 | 32.92-70.36 | 1929- | - |
| Lagunitas** | AMC | 2765 | 33.07-70.26 | 1959-2014 | 1974-2014 |
| Embalse el Yeso** | DGA | 2475 | 33.68-70.09 | 1962- | 1962- |
| Cristo Redentor* | SMN | 3829 | 32.83 70.06 | 1965-1984 | 1956-1985 |
| Horcones* | SMN | 2988 | 32.81-69.94 | 1955- | 1999- |
| Puente del Inca* | SMN | 2733 | 32.82-69.90 | 1956-1976 | 1956-1976 |
| Punta de Vacas | SMN | 2441 | 32.88-69.77 | 1992- | 1992- |
| Punta de Vacas II* | SMN | 2405 | 32.85-69.76 | 1998-2007 | 1998-2007 |
| Polvaredas | SMN | 2249 | 32.79-69.65 | 1983- | - |
| East I zone | | | | | |
| San Alberto | SMN | 2190 | 32.47-69.41 | 1983- | - |
| Uspallata II | SMN | 1885 | 32.60-69.35 | 1962-2014 | 1962-2014 |
| Uspallata | SMN | 1896 | 32.59-69.34 | 1983- | 1993- |
| East II zone | | | | | |
| Guido | SMN | 1418 | 32.92-69.24 | 1957- | 1965- |
| Potrerillos | SMN | 1448 | 32.96-69.20 | 1983- | - |
| Cacheuta** | SMN | 1270 | 33.01-69.12 | 1983- | - |
| Cerro Pelado* | SMN | 3047 | 32.76-69.10 | 1983- | - |


**Table 2:** Decadal changes in the precipitation percentage (Pp), the *number of days* with snowfall-driven G-CHs (S) and the *number of days* with rainfall-driven G-CHs (R). The values with asterisks are significant at 95 %, the empty spaces correspond to data not available.

| | 1957-2017 | | | | | | 1961-2017 | | | | | |
|---|---|---|---|---|---|---|---|---|---|---|---|---|
| | Summertime | | | Wintertime | | | Summertime | | | Wintertime | | |
| | Pp | S | R | Pp | S | R | Pp | S | R | Pp | S | R |
| **West** | 1.07 | | | -1.07 | | | 0.90 | | 0.00 | -3.91 | 0.00 | 0.00 |
| **East** | 5.12 | | | 6.70* | | | 4.53 | | 0.20* | 8.46* | 0.00 | 0.00 |




**Figure 1: Main features of the study area and G-CHs. (a) Location of the study area (inset) and number of G-CHs for each ravine, talus cone and rock wall monitored superimposed on a topographic map. (b) Topographic profile along the Mendoza river. (c) G-CHs number for each trigger and zone. The yellow line in panels (a) and (b) delimits zones W and E.**





**Figure 2: Mean annual cycles of selected variables. (a) Monthly mean standardized precipitation along a west-east transect. The vertical axis indicates the distance eastward from the Andes ridge. Diamonds indicate the stations positions, vertical lines the seasonal division used and horizontal thick (thin) line the longitudinal division used (commented). (b) periods of prevailing precipitation by frontal systems in each zone. (c) range of monthly mean standardized precipitation considering the stations in each zone. (d) mean average standardized temperature in each zone. In panels (b), (c) and (d) yellow (grey) lines refers to the W (E) zone.**

**Figure 3: Pictures illustrating G-CHs in study area: (a,c) rainfall-driven debris flow occurred on 2 February 2016 in the E zone, (b,d) rainfall-driven debris flow occurred on 23 January 2016 in the E zone, (e) rainfall-driven debris flow occurred on 4 February 2018 in the E zone, (f) rainfall-driven debris falls occurred on 2 August 2013 in the E zone, (g,h) snow avalanches in the W zone.**





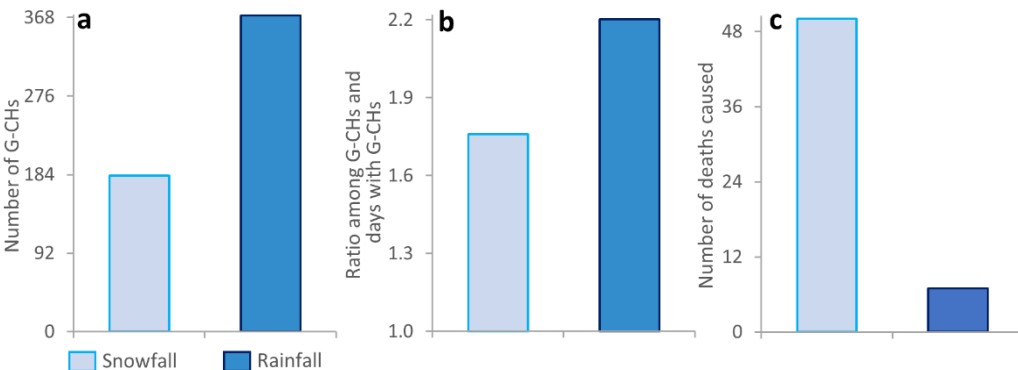

**Figure 4: Main statistics of the G-CHs triggers considering the entire record. (a) Total number, (b) relationship between *number of*** 
***G-CHs* and *number of days* with G-CHs, (c) associated fatalities.**

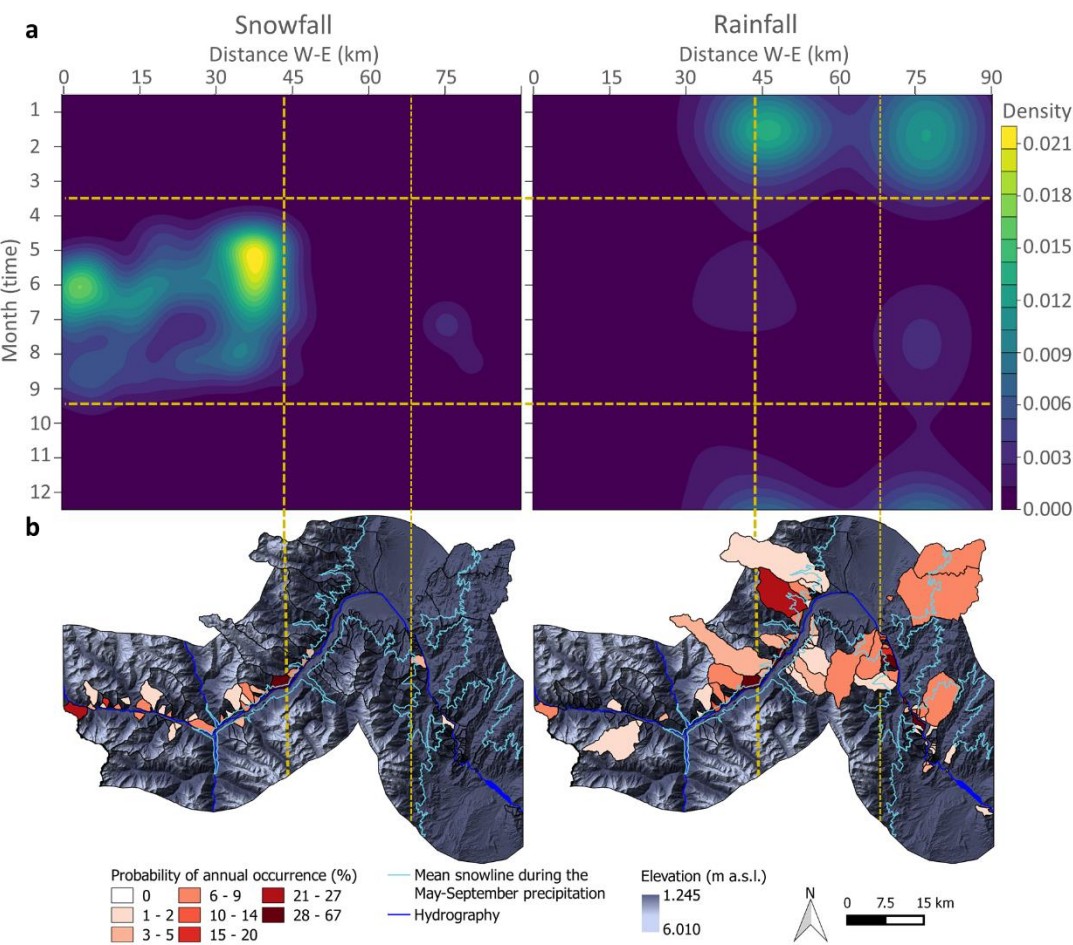

**Figure 5: Distributions of the two triggers of G-CHs. (a) Temporal probability density along a west-east transect for snowfall (left panel) and rainfall (right panel). The horizontal axis indicates the distance eastward from the Andes ridge. (b) Probability of G-CHs annual occurrence for each ravine, talus cone and rock wall monitored superimposed on a topographic map. The cyan lines**
**indicate the mean winter freezing level. Horizontal lines indicate the seasonal division used and vertical thick (thin) line the longitudinal division used (commented).**




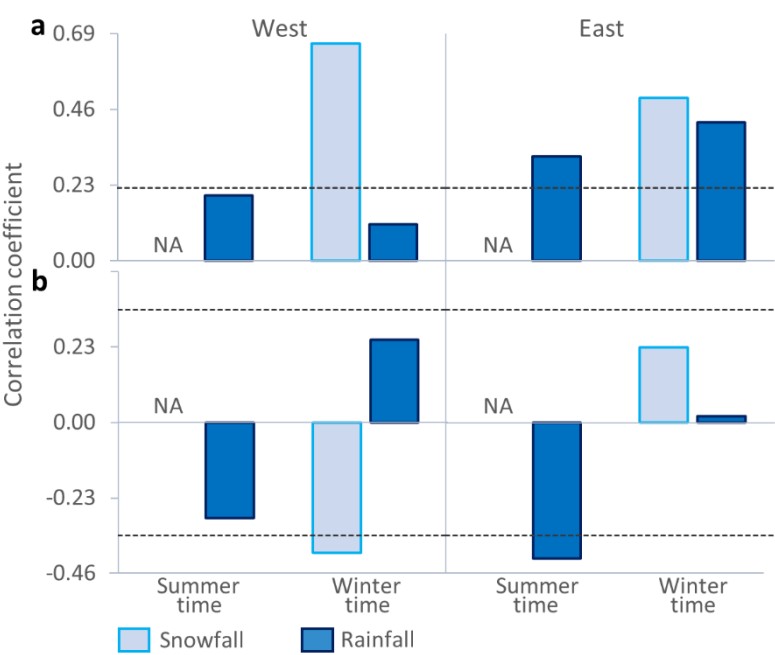

**Figure 6: Inter-annual correlations between the series of the *number of days* with G-CHs per season and: (a) precipitation and (b) mean temperature. Dashed lines indicate significance at 95 % level.**



**Figure 7: Trends in drivers and G-CHs.** Series of precipitation (grey lines) and *number of days* with G-CHs (lines with points) in the W zone (left column) and E zone (right column) for: (a-d) summer and (e-h) winter months. In the case of G-CHs, blue colour refers to rainfall-driven hazards and light-blue to snowfall-driven hazards. Dashed lines indicate significant trends at 95 % level.



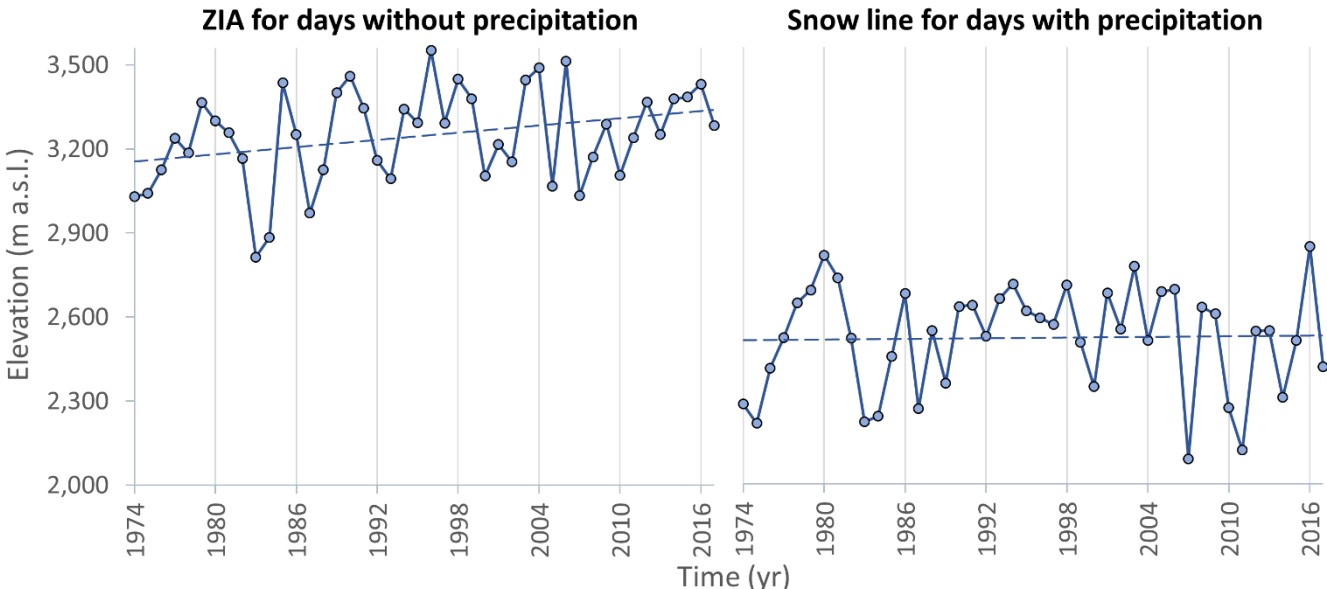

**Figure 8: Warming effects. Left panel: annual series for the May-September average of the Zero-degree isotherm altitude during dry days. Right panel: annual series for the May-September average of the snowline during days with precipitation. Dashed lines indicate the trends.**


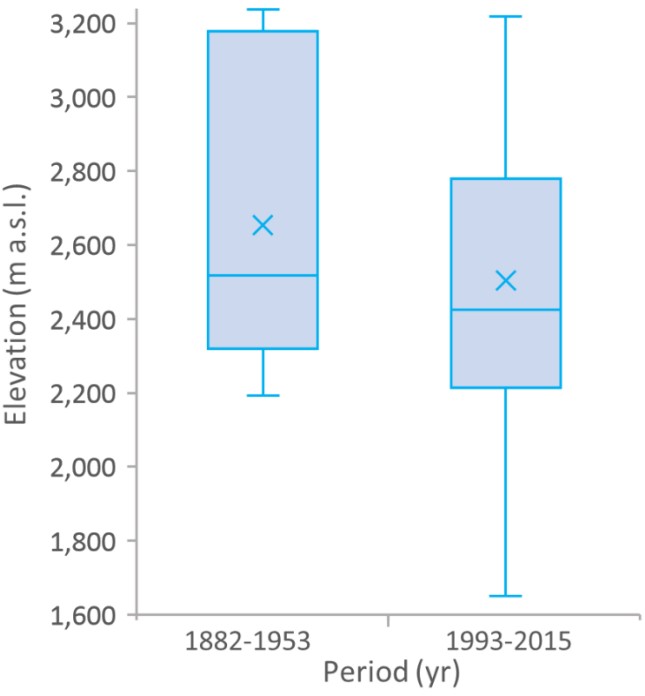

**Figure 9: Boxplots for the elevations of the main deposits of snowfall-driven G-CHs during the periods 1882-1953 and 1993-2015.**