# Peer review of "Geo-climatic hazards in the eastern subtropical Andes: Distribution, Climate Drivers and Trends"

_Natural Hazards and Earth System Sciences, 2019_

## Referee Comment (RC1) · Anonymous Referee #1 · 3 Feb 2020

This manuscript offers a clear and interesting analysis of the interaction of snowfall and rainfall on the triggering of landslides and snow avalanches in a section of the Argentinean central Andes, where they pose a significant hazard and risk in a very busy transport corridor. The hazards are divided in two zones with different climatic patterns, which allow statistical analysis to assess the effects of climate and global warming on the hydrometeorological hazards activity. The manuscript is well written, figures are fine and results are sound.

My main comments are on the line of providing some more detail in the statistical methods used for the analyses, rather tan just giving a reference citation, and in particular to provide, if possible, or at least comment in the discussion, more detailed insights on the relationship of snowfall and rainfall patterns with landslide types. You have a

database with landslides classified as debris flows, falls, rotational and translational slides and complex landslides, with nearly 80% of them debris flows. Is it possible to get relationships for those types as separate subsets of data? They may be not statistically significant, but it would be interesting to comment on this. Are the results biased for debris flows?, thus are they applicable for the other landslide types? A second issue is that for the analysis you separate the area into terrain units of ravine, talus and rock walls, which are very variable in size. Could you explain the criteria to define these terrain units, which are used for probability assessment? Is the size difference a problem? I presume they are linked to some preferent landslide type (e.g. rock wall for falls, ravine for flows), is then possible to analyse the data in subsets of landslide type and/or terrain unit?

Some minor comments on the manuscript are the following:

L55-57. What can you say about g-CHs in the Chilean side? Are they absent, or there are no data?

L67-79 In this paragraph you provide some details on debris flows characteristics and mechanics, but say nothing on the other landslide types or snow avalanches also included in the analysis, can you homogeneize the information?

L144-147 could you mention the proportion of those landslides/avalanches triggered by earthquakes or other identified triggers in comparison with climatic or unknown trigger?

L173-174 could you please explain a bit these methods in the Methods chapter? Computing the probability is of the the most significant aspects for hazard analysis.

L220-227 Please explain why you use surface temperature data only from the Chilean side? are they representative?

L465 please revise the sentence "Horizontal lines indicate the seasonal division used and vertical thick (thin) line."

---

## Referee Comment (RC2) · Martin Mergili (Referee) · 5 Feb 2020

The authors try to correlate the spatio-temporal occurrence patterns of geo-climatic hazards (G-CHs) in the Andes near Mendoza, western Argentina, to precipitation and temperature patterns. Thereby, they focus on snow avalanches and landslides. The study presents a highly relevant topic, and the manuscript is generally concisely and well written, structured, and illustrated. Not all statistical analyses yield significant results, but this is clearly communicated, so that it is possible to capture the essence of the outcomes. I have some recommendations to the authors which should be considered before publication. Consequently, I suggest minor revisions.

- Even though the paper is well understandable, there are some issues of grammar

and style. Improvements should be included in the revised version, even though final polishing can be done through copy-editing anyway.

- The 15 km buffer on both sides of the Río Mendoza appears a bit arbitrary to me. Is there a specific reason why exactly this distance was used? And might it be an alternative way to consider those catchments draining to the Mendoza Valley, instead of a fixed distance? This would probably not change the results at all – but the maps (which are in principle very nice) look a bit awkward with the buffer of the elevation map and the catchments with G-CHs reaching beyond that buffer.

- In Line 121, it is mentioned that the G-CHs are concentrated in ravines, talus cones, and rock walls. I suppose that the individual zones were derived by computing the catchment areas (particularly for the ravines). This is absolutely fine, but you should mention it explicitly.

- It would be interesting to know a little bit more about how reliably you could determine the occurrence of daily precipitation. You mention the use of the CMORPH data. Did this work well? And how did you do it before 2002? Particularly convective events can have a very patchy occurrence, and are not necessarily recorded at stations. How did you deal with this issue, and do you expect that it significantly influences your results?

- Table 1: some of the meteo stations do not cover the entire investigation period – could this induce some bias in the derived trends? You might wish to briefly discuss this issue.

- Fig. 1: maybe you could put the names of the major settlements, or label the meteorological stations. Some labelling would be nice in this introductory map.

- A minor issue in the legend of Fig. 5: the probability of annual occurrence is a continuous number. This should also be reflected in the legend – as it is now, e.g. a probability of 2.5% would not be covered at all. Better write: 0; >0-2; >2-5; ... etc.

2019-381, 2020.

---

## Author Response (AR1)

**RESPONSE**-Referee #1

 **General Comments**

*This manuscript offers a clear and interesting analysis of the interaction of snowfall and rainfall on the triggering of landslides and snow avalanches in a section of the Argentinean central Andes, where they pose a significant hazard and risk in a very busy transport corridor. The hazards are divided in two zones with different climatic patterns, which allow statistical analysis to assess the effects of climate and global warming on the*
10 *hydrometeorological hazards activity. The manuscript is well written, figures are fine and results are sound.*

Thank you for your general and specific comments. They have helped to improve the manuscript. We addressed all the observations and below are our point-by-point responses.

*My main comments are on the line of providing some more detail in the statistical methods used for the*
15 *analyses, rather than just giving a reference citation, and in particular to provide, if possible, or at least comment in the discussion, more detailed insights on the relationship of snowfall and rainfall patterns with landslide types.*

Thanks for raising this point. Where necessary we clarified the statistical methods (lines 191-193) and we expanded our discussion on the G-CH types and climate drivers (lines 299-301).

*You have a database with landslides classified as debris flows, falls, rotational and translational slides and complex landslides, with nearly 80% of them debris flows. Is it possible to get relationships for those types as separate subsets of data? They may be not statistically significant, but it would be interesting to comment on this. Are the results biased for debris flows?, thus are they applicable for the other landslide types?*

25 In the new historical record of events, there is no a dominant type of G-CH: 31% are flows, 33% are falls, 20% are snow avalanches and 16% are undetermined (lines 156-162). Thus, while your suggestion is interesting we have refrained from getting specific relationships for each subset given the small number of events in those samples. The aggregated analysis, on the other hand, provides more statistical robustness. Due to the rather even distribution of G-CHs types we don't think that
30 results are biased for debris flows. We have commented on this issue in lines 178-180.

*A second issue is that for the analysis you separate the area into terrain units of ravine, talus and rock walls, which are very variable in size. Could you explain the criteria to define these terrain units, which are used for probability assessment? Is the size difference a problem? I presume they are linked to some preferent landslide*

35 *type (e.g. rock wall for falls, ravine for flows), is then possible to analyse the data in subsets of landslide type and/or terrain unit?*

The definition of the terrain units is now described in section 2 (lines 128-129). All terrain units that intercept the route or the railway were drawn using hydrological tools of the SAGA software. The G-CHs were assigned to the different units either because the sources indicate the name of the activated
40 ravine or the kilometre of the route or the railway that were cut. For the second case, the distances along the route and the railway were georeferenced to know which terrain units were activated. Sometimes talus cones and rock walls (not always automatically separable with the channel network) present activities in specific sectors of a terrain unit. In these cases, the terrain units were subdivided but having a minimum area limit of 0.2 km$^2$, in order to not delimit a terrain unit in each place where a
45 fall or a debris avalanche occurred.
The different size of the terrain units was not a problem for the spatio-temporal probability assessment since we used the date of each G-CH and the along-route distance that was affected by the G-CH. We are aware that having the volume of each G-CH or a proxy of this could have resulted in a better probability assessment, but such volume is unknown for the vast majority of the cases.

50
**Minor Points**

*L55-57. What can you say about g-CHs in the Chilean side? Are they absent, or there are no data?*

In the valley of the Aconcagua River (Chilean side of the international road) there are also many G-CHs (e.g., Sepúlveda and Moreiras, 2013; Sepúlveda et al., 2015) with negative impacts in infrastructure
55 and transportation. In this work, however, we have focused on the Argentinean side give the long, high quality record developed for this sector. Note that the route and the railroad change the jurisdiction, and the databases of these organisms, which are the most important source, were not available for the Chilean side. We acknowledge this limitation in lines 55-57.

60 *L67-79 In this paragraph you provide some details on debris flows characteristics and mechanics, but say nothing on the other landslide types or snow avalanches also included in the analysis, can you homogeneize the information?*

Thanks for the comment. We add information about the other landslide types and the snow avalanches in lines 75-77.

65

*L144-147. could you mention the proportion of those landslides/avalanches triggered by earthquakes or other identified triggers in comparison with climatic or unknown trigger?*

59 landslides triggered by earthquakes (9% of the total landslides and snow avalanches), 16 landslides by snowmelt (2% of the total) and 55 landslides without an established trigger (8% of the total) were
70 counted. Information added in lines 52-54.

*L173-174 could you please explain a bit these methods in the Methods chapter? Computing the probability is of the most significant aspects for hazard analysis.*

Thanks for the comment, information about the method to calculate the probability was added in lines 191-193.

75

*L220-227. Please explain why you use surface temperature data only from the Chilean side? are they representative?*

These high-elevation station are located at 25-80 km from W zone where 87% of the snowfall-driven G-CHs take place. These distances are acceptable for the typical spatial variation of temperature. It is
80  now indicated in the text because these stations were used and not those in Argentina (lines 248-250).

*L465 please revise the sentence "Horizontal lines indicate the seasonal division used and vertical thick (thin) line."*

We have corrected this phrase (lines 509-510).

85

**New References**

Sepúlveda, S. A., and Moreiras, S. M.: Large volume Landslides in the Central Andes of Chile and Argentina (32°-34°S) and related hazards, Bulletin of Engineering Geology and the Environment, 6,
90  287-294, 2013.

Sepúlveda, S. A., Moreiras, S. M., Lara, M., and Alfaro, A.: Debris flows in the Andean ranges of central Chile and Argentina triggered by 2013 summer storms: characteristics and consequences, Landslides, 12(1), 115-133, 2015.

95

100

**RESPONSE-Referee #2 (Martin Mergili)**

**General Comments**

*The authors try to correlate the spatio-temporal occurrence patterns of geo-climatic hazards (G-CHs) in the Andes near Mendoza, western Argentina, to precipitation and temperature patterns. Thereby, they focus on snow avalanches and landslides. The study presents a highly relevant topic, and the manuscript is generally concisely and well written, structured, and illustrated. Not all statistical analyses yield significant results, but this is clearly communicated, so that it is possible to capture the essence of the outcomes. I have some recommendations to the authors which should be considered before publication. Consequently, I suggest minor revisions.*

[Reply] We thank and appreciate your general and specific comments. Your inputs helped us to improve the manuscript. We addressed all the observations made as you can see below.

*Even though the paper is well understandable, there are some issues of grammar and style. Improvements should be included in the revised version, even though final polishing can be done through copy-editing anyway*

[Reply] Done. We polished the revised version of our manuscript.

*The 15 km buffer on both sides of the Río Mendoza appears a bit arbitrary to me. Is there a specific reason why exactly this distance was used? And might it be an alternative way to consider those catchments draining to the Mendoza Valley, instead of a fixed distance? This would probably not change the results at all – but the maps (which are in principle very nice) look a bit awkward with the buffer of the elevation map and the catchments with G-CHs reaching beyond that buffer.*

[Reply] The buffer was chosen to show the minimum, average and maximum elevations of the area (Fig. 1b) and to give an average elevation value of the two zones that we used in our study. Therefore, its utility is secondary. The alternative of using a larger rectangle covering the whole region for Fig. 1a results in a very similar map.

*In Line 121, it is mentioned that the G-CHs are concentrated in ravines, talus cones, and rock walls. I suppose that the individual zones were derived by computing the catchment areas (particularly for the ravines). This is absolutely fine, but you should mention it explicitly.*

[Reply] Done... We now mention how we draw the terrain units in lines 128-129.

*It would be interesting to know a little bit more about how reliably you could determine the occurrence of daily precipitation. You mention the use of the CMORPH data. Did this work well? And how did you do it before 2002? Particularly convective events can have a very patchy occurrence, and are not necessarily recorded at stations. How did you deal with this issue, and do you expect that it significantly influences your results?*

140 [Reply] Assessing the occurrence (set aside the amount) of precipitation over the Andes is indeed a major challenge. The task is more simple in wintertime, when there is a good agreement between CMORPH and station data given the widespread nature of frontal precipitation (yet, how much snow fall in each storm is largely unknown). But as you guessed, the situation is more complex in summertime, when rainfall is delivered by convective systems. To assign the *rain* trigger to the G-CHs

145 that occurred during the convective season and for which the source did not inform the trigger, first it was seen if potentially influential earthquakes had occurred (Moreiras et al., 2006) or if conditions for rapid melting were met (Vergara et al., 2020). Only then it was observed if some of the nearby stations or the CMORPH had measured some precipitation. We acknowledge that identification of summertime rainfall-induced hazards have higher uncertainty that winter G-CHs and discuss further on this in lines

150 148-151.

*Table 1: some of the meteo stations do not cover the entire investigation period –could this induce some bias in the derived trends? You might wish to briefly discuss this issue.*

[Reply] Effectively, not taking cautions in the combination of stations with unequal record periods

155 would lead to errors in the calculation of trends. In this work precautions were taken. This procedure is better explained now (lines 205-211).

*Fig. 1: maybe you could put the names of the major settlements, or label the meteorological stations. Some labelling would be nice in this introductory map.*

160 [Reply] The names of the stations could not be added since they are 16 and there is little free space. But the locations of some towns were added.

*A minor issue in the legend of Fig. 5: the probability of annual occurrence is a continuous number. This should also be reflected in the legend – as it is now, e.g. a probability of 2.5% would not be covered at all. Better write:*

165 *0; >0-2; >2-5;*

[Reply] Thank you for warning us of this error, now it was corrected.

170

**New References**

[revised manuscript text omitted]

The purpose of this research is to assess the significance of the G-CHs trends and investigate their causes. To attribute these trends, we began by improving our general understanding of regional G-CHs analysing their distributions in time and space, and establishing their relationship with different climate elements. This research took advantage of a long, updated record of G-CHs in the eastern side of the subtropical Andes, with daily resolution and precise spatial location along a portion of an international, highly transited highway connecting Chile and Argentina. On the other hand, meteorological information is rather poor in this area, with few surface stations and absence of other measurement systems (radar, local radiosondes, etc.). The paper is organized as follows. We begin by describing the physical characteristics of the area, the assembly of the G-CHs record and its usage. In the results section we present the spatio-temporal characterization of the G-CHs, followed by their association with the local climate and the temporary changes of the climatic variables and the G-CHs. In the discussion section, the results obtained are analysed, a future research direction is proposed, and a conceptual evaluation of the future geo-climatic hazard of the region is carried out. Finally, in the conclusions section, the most important results are described.

**1.1 Study area and geographical setting**

At subtropical latitudes (32-33° S) the Andes cordillera separates central Chile (to the west) and western central Argentina (to the east) with the border approximately following the highest peaks of the range, that reaches over 5 km ASL in this sector (Fig. 1). Here, an international road and the ex-Trasandino railway, links the cities of Los Andes (Chile) and Mendoza (Argentina). This is a major commercial and touristic route in use since the 17th century connecting the east and west side of the continent. G-CHs occur at both sides of the Andes (e.g., Sepúlveda and Moreiras, 2013; Sepúlveda et al., 2015) but here we focus on the eastern side because of the availability of a historical record taken in the Argentinean side of the road and railway, extending eastward from the border down to the plains near the city of Mendoza. This sector coincides with the middle and upper Mendoza river basin (Fig. 1a) and hosts more than 10 thousand inhabitants. Anthropic changes in the landscape are negligible due to the absence of significant cultivated areas and the limited infrastructure. The National Route 7 has currently an average traffic of about 3000 vehicles per day (ONDaT, 2018), maintained throughout the year due to domestic travels and the daily opening of the international pass, except when large snowfalls or major G-CHs occur.

The study area has elevations from 1250 to 5970 m ASL (when using a buffer of 15 km with respect to the Mendoza River that roughly encompassed all the terrain units studied; Fig. 1b). The lower limit of discontinuous permafrost is at 3700 m ASL approximately (Trombotto et al., 1997). The area covered by perennial snow and glaciers (including inactive rock glaciers) is 103 km$^2$ (2.7 % of the total area; IANIGLA, 2018). The study area encompasses the morphotectonic units

[revised manuscript text omitted]